# DAGPrompT: Pushing the Limits of Graph Prompting with a Distribution-aware Graph Prompt Tuning Approach

## Abstract

The "pre-training then fine-tuning" paradigm has advanced Graph Neural Networks (GNNs) by enabling the capture of general knowledge without task-specific labels. However, a significant objective gap between pre-training and downstream tasks limits their effectiveness. Recent graph prompting methods aim to bridge this gap by task reformulations and learnable prompts. Yet, they struggle with complex graphs like heterophily graphs—freezing the GNN encoder may diminish prompting effectiveness, and simple prompts fail to capture diverse hop-level distributions. This paper identifies two key challenges in adapting graph prompting methods for complex graphs: (i) *adapting the model to new distributions in downstream tasks to mitigate pre-training and fine-tuning discrepancies from heterophily* and (ii) *customizing prompts for hop-specific node requirements.* To overcome these challenges, we propose Distribution-aware Graph Prompt Tuning (DAGPrompT), which integrates a GLoRA module for optimizing the GNN encoder's projection matrix and message-passing schema through low-rank adaptation. DAGPrompT also incorporates hop-specific prompts accounting for varying graph structures and distributions among hops. Evaluations on 10 datasets and 14 baselines demonstrate that DAGPrompT improves accuracy by up to 7.55% in node and graph classification tasks, setting a new state-of-the-art while preserving efficiency. We provide our code and data via AnonymousGithub.

## CCS Concepts

• **Mathematics of computing** → **Graph algorithms**; • **Computing methodologies** → *Neural networks*; Supervised learning.

## Keywords

graph neural networks, graph prompting, few-shot learning

**ACM Reference Format:**
Anonymous Author(s). 2018. DAGPrompT: Pushing the Limits of Graph Prompting with a Distribution-aware Graph Prompt Tuning Approach. In *Proceedings of Make sure to enter the correct conference title from your rights confirmation emai (Conference acronym 'XX).* ACM, New York, NY, USA, 13 pages. https://doi.org/XXXXXXX.XXXXXXX

## 1 Introduction

In recent years, the schema of "pre-training then fine-tuning" on Graph Neural Networks (GNNs) has experienced significant growth,

especially in few-shot learning scenarios [12, 19, 30, 44]. Specifically, GNNs are pre-trained in a self-supervised manner on tasks such as graph property reconstruction [10, 14] or contrastive learning [26, 40]. Then, GNNs are adapted to downstream tasks during the fine-tuning. However, a common limitation is that the gap between pre-training and downstream objectives is often overlooked, which hinders the model performance. For example, the pre-training objective may be link prediction, while the downstream objective may be node classification, and these two objectives vary a lot [28]. To address this, recent research has begun incorporating graph prompting techniques [9, 18, 27, 28] to bridge the gap between pre-training and downstream tasks. They propose using prompts to reformulate downstream tasks as pre-training tasks with additional learnable parameters. The pre-trained GNN encoder remains frozen during this process. For example, GPPT [27] reformulates the downstream task, node-classification, to the pre-training task, link-prediction. This reformulation reduces the objective gap between the pre-train and downstream by the alignment of objective forms.

However, existing prompting methods are sub-optimal for graphs with complex distributions, such as heterophily graphs, where connected nodes frequently have different labels [6, 42]. *This label disparity creates a profound disconnect between pre-training objectives and downstream tasks.* As most pre-training techniques are label-agnostic and rely on graph structure to varying extents, they inherently suffer from this discrepancy. For instance, tasks like link prediction push the model to generate similar embeddings for connected nodes, ignoring label differences. Consequently, connected nodes with distinct labels are mapped to similar embeddings in heterophily graphs, as shown in Figure 1. During prompting, current approaches [9, 18, 27, 28, 37, 41] typically freeze the GNN encoder and employ basic prompting techniques (e.g., projection or additive layers). However, freezing the GNN encoder restricts its adaptability to distribution shifts in downstream tasks. As illustrated in Figure 1, this limitation prevents the model from adjusting GNN parameters to produce distinct node embeddings for different labels. The basic prompting mechanisms struggle to disentangle node embeddings effectively, ultimately leading to reduced performance.

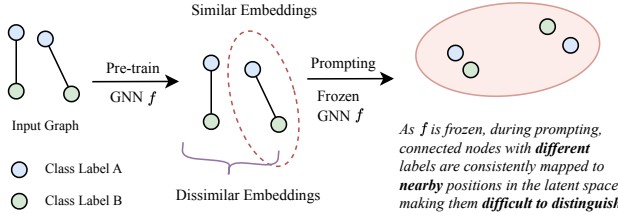

**Figure 1: Heterophily diminishes the effectiveness of prompting techniques that freeze the GNN encoder, resulting in indistinguishable node embeddings.**

Moreover, distributional differences across graph hops in complex graphs further challenge existing prompting methods: some nodes prioritize information from closer hops, while others rely more on distant ones [33, 42]. Additionally, the heterophily distribution varies across hops (see Figure 7 in Appendix D) and affects performance to varying extents. Existing methods typically rely on the final layer representation of the GNN encoder, overlooking these hop-specific variations [18, 27, 41]. While some approaches [9, 28] attempt to merge intermediate embeddings, they fail to account for hop-specific preferences or offer tailored prompts for different hops, limiting their adaptability to these variations.

In summary, two primary challenges arise when extending graph prompting methods to diverse graph data with varying internal distributions: (i) *adapting the model to new distributions in downstream tasks, reducing the discrepancy between pre-training and fine-tuning caused by heterophily*, and (ii) *customizing prompts to address the hop-specific requirements of different nodes*. These challenges stem from the inherent limitations of the prompting paradigm and the complexities of heterophily graphs, making them critical obstacles in advancing graph prompting techniques. While existing methods perform well on homophily graphs, they degrade on heterophily graphs due to neglecting these issues. We compare the 5-shot node classification accuracy of a popular prompting method, GPPT [27], with a GCN trained from scratch [13] and pre-trained models using link prediction [19] and DGI [30], as shown in Figure 2. GPPT and pre-training strategies perform well on the homophily graph Cora but are outperformed by a GCN trained from scratch on the heterophily graphs Texas and Cornell, which exhibit strong heterophily and variation across hops (see Figure 7 in Appendix D).

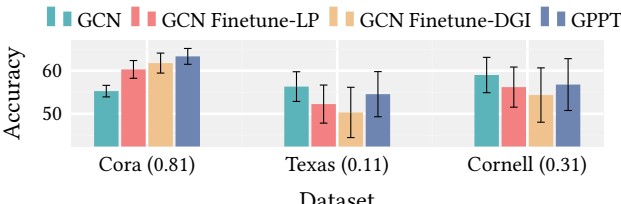

**Figure 2: Conventional graph prompting techniques are less effective or even detrimental on heterophily datasets. The homophily ratio [42] is indicated in the brackets, with a lower ratio representing stronger heterophily.**

In this paper, we propose **Distribution-aware Graph Prompt Tuning (DAGPromT)**, which comprises two core components in response to the two challenges outlined above: (i) *Graph Low-Rank Adaptation (GLoRA)* module: This module leverages low-rank matrix approximations to tune both the projection parameters and the message-passing mechanism in the GNN encoder. By doing so, it adapts the GNN encoder to the new distributions encountered in downstream tasks, while preserving the valuable knowledge embedded in pre-trained weights in an efficient manner. For example, in the scenario depicted in Figure 1, GLoRA addresses the limitations by enabling the GNN encoder to produce separable embeddings for nodes with different labels. (ii) *Hop-specific Graph Prompting* module: This module decomposes downstream tasks

into hop-specific components, allowing the model to weigh the importance of different hops adaptively. We validate DAGPromT through experiments on 10 datasets, focusing on both node and graph classification tasks, and comparing it against 14 baseline methods. Our model achieves state-of-the-art performance, with an average accuracy improvement of 3.63%, and up to 7.55%. In summary, our contributions are as follows:

- We identify the key challenges of applying graph prompting techniques to heterophily graphs and introduce DAGPromT as a solution. DAGPromT distinguishes itself as a pioneering model in extending the capabilities of graph prompting techniques to heterophily graphs.
- We propose two key modules for DAGPromT: GLoRA and the Hop-specific Graph Prompting module. These modules mitigate distributional misalignment between pre-training and downstream tasks, and adapt the model to the diverse distributions across hops.
- We conduct extensive experiments on both node and graph classification tasks using 10 datasets and 14 baselines. DAGPromT demonstrates remarkable performance improvements, improving the accuracy up to 7.55%. We further evaluate DAGPromT in terms of data heterophily level, number of shots, transferability, efficiency, and ablation studies.

## 2 Preliminary

*Notations.* Consider an undirected graph $\mathcal{G} = \{\mathcal{V}, \mathcal{E}\}$, where $\mathcal{V}$ represents the set of $N$ nodes and $\mathcal{E}$ represents the set of $E$ edges. The graph is described by its adjacency matrix $\mathbf{A} \in \mathbb{R}^{N \times N}$, where $\mathbf{A}_{ij} = 1$ if and only if there exists an edge $e_{ij} \in \mathcal{E}$ connecting node $v_i$ and node $v_j$. Additionally, each node $v_i \in \mathcal{V}$ is associated with the feature vector $\mathbf{X}_i \in \mathbb{R}^{N \times F}$ and a label $\mathbf{y}_i$, with $F$ representing the dimension of the node features.

*Graph Homophily Measurements.* Real-world graphs are inherently complex, often featuring diverse internal structures where nodes follow varying patterns [20, 24]. Heterophily provides a useful lens for analyzing these structures, particularly through the concept of label consistency, measured by the homophily ratio [42] $h = \frac{|\{(v_i, v_j) : (v_i, v_j) \in \mathcal{E} \wedge \mathbf{y}_i = \mathbf{y}_j\}|}{|\mathcal{E}|}$. $h$ represents the fraction of edges in $\mathcal{E}$ that connect nodes with the same label. High homophily indicates strong similarity between connected nodes ($h$ near 1), while low homophily suggests greater dissimilarity ($h$ near 0).

*Graph Neural Networks (GNNs).* The remarkable success of GNNs can largely be attributed to the message-passing mechanism [32], and a general GNN layer can be defined as:

$$\mathbf{H}_i^{(l+1)} = f_\theta \left( \text{AGGR} \left( \mathbf{H}_i^{(l)}, \left\{ \mathbf{H}_j^{(l)} : v_j \in \mathcal{N}_i \right\} \right) \right), \quad (1)$$

where $\mathbf{H}^{(l)} \in \mathbb{R}^{N \times d^{(l)}}$ represents the node embeddings at layer $l$, with initial embeddings $\mathbf{H}^{(0)} = \mathbf{X}$. The term $d^{(l)}$ denotes the dimensionality at layer $l$, AGGR$(\cdot)$ aggregates the neighboring node embeddings, and $f_\theta$ applies a projection, for example, using a linear transformation followed by non-linear activation (e.g., ReLU).

*Fine-tuning Pre-trained GNNs.* For a pre-trained GNN model $f$, a learnable projection head $\theta$, and a downstream task (e.g., node

Figure 3: The framework of Distribution-aware Graph Prompt Tuning.

classification) dataset $\mathcal{D}$, we fine-tune the parameters of $f$ and $\theta$ to maximize the likelihood of predicting the correct labels $\mathbf{y}$ in $\mathcal{D}$:

$$\max_{f,\theta} P_{f,\theta}(\mathbf{y} \mid \mathcal{D}) \qquad (2)$$

*Prompting Pre-trained GNNs.* Given a GNN model $f$ pre-trained under task $\mathcal{T}_{pt}$, a set of learnable prompting parameters $\{\theta\}$, and a downstream task dataset $\mathcal{D}$ for task $\mathcal{T}_{ds}$, we introduce a task reformulation function $h$. This function maps the downstream task to a form consistent with the pre-trained task $\mathcal{T}_{pt}$. For instance, node classification ($\mathcal{T}_{ds}$) can be reformulated as link prediction ($\mathcal{T}_{pt}$) by introducing pseudo nodes, where labels are assigned by predicting the most probable links between nodes and pseudo nodes [27]. During the prompting, the parameters of $f$ are frozen, and we optimize $\theta$ to maximize the likelihood of predicting the correct labels $\mathbf{y}$, guided by $h$:

$$\max_{\{\theta\}} P_{f,\{\theta\}}(\mathbf{y} \mid h(\mathcal{D})) \qquad (3)$$

## 3 Method

In this section, we elaborate on the Distribution-aware Graph Prompt Tuning (DAGPrompT). The framework of DAGPrompT is illustrated in Figure 3, which consists of two stages: (i) Link-prediction-based pre-training. (ii) Graph Low-rank Adaptation with Hop-specific Graph Prompting. We also provide a detailed algorithm in Appendix A and a complexity analysis in Appendix B.

### 3.1 Label-free Pre-Training

The pre-training strategy is essential for few-shot learning, allowing the model to capture graph structures across diverse domains without labeled data, as shown by several approaches [18, 27, 28, 41]. It also aids in capturing local structures and reduces over-fitting [19]. We adopt link prediction for pre-training due to its advantages: (i) the abundance of inherent edge data in graphs, and (ii) alignment in objective forms between pre-training and downstream tasks, as tasks like node and graph classification can be seamlessly reformulated as link prediction by introducing pseudo-nodes or pseudo-graphs [18, 27].

Consider a node $v$ in a graph $\mathcal{G}$. For training, a positive node $a$ is selected from the neighbors of $v$, and a negative node $b$ from the non-neighbors, forming a triplet $(v, a, b)$. Let the GNN encoder $f$ produce the corresponding embeddings $\mathbf{s}_v$, $\mathbf{s}_a$, and $\mathbf{s}_b$. By considering all nodes in $\mathcal{G}$, the pre-training dataset $\mathcal{T}_{pt}$ is constructed. The pre-training loss is then defined as:

$$\mathcal{L}_{pt}(\Theta) = - \sum_{(v,a,b) \in \mathcal{T}} \ln \frac{\exp\left(\text{sim}\left(\mathbf{s}_v, \mathbf{s}_a\right)/\tau\right)}{\sum_{u \in \{a,b\}} \exp\left(\text{sim}\left(\mathbf{s}_v, \mathbf{s}_u\right)/\tau\right)}, \qquad (4)$$

where $\tau$ is a temperature hyper-parameter that controls the sharpness of the output distribution, and $\Theta$ represents the parameters of the function $f$. The goal of pre-training for link prediction is to push node embeddings connected by edges closer in the latent space, while separating those without connections [18].

### 3.2 Distribution-aware Graph Prompt Tuning

In this subsection, we discuss tuning and prompting the pre-trained GNN $f$ for downstream tasks in a distribution-aware approach. We introduce the Graph Low-Rank Adaptation (GLoRA) module, which aligns the projection and message passing scheme of $f$ with the distribution of downstream tasks through low-rank adaptation. This approach preserves the knowledge embedded in the pre-trained weights while adapting to new tasks. We then detail the prompting module, which links diverse downstream tasks to the pre-training objective, ensuring alignment with the unique downstream distributions in a hop-decoupled manner.

*3.2.1 Tuning with Graph Low-Rank Adaptation.* Previous works often pre-train GNNs and keep them frozen during prompting, relying on learnable prompts for downstream tasks [9, 18, 27, 28]. While effective in graphs with strong homophily, this approach underperforms in more complex settings, such as graphs with strong heterophily, as shown in Figure 2. Freezing the GNN can lead to performance degradation in such cases, as most pre-training methods are label-agnostic, and downstream objectives often differ from pre-training goals, especially in heterophily graphs. For instance, link prediction favors similar embeddings for connected nodes, which aligns with homophily but fails in heterophily, where connected nodes may have dissimilar characteristics. As illustrated in Figure 1,

on heterophily graphs, pre-training without tuning the GNN encoder can result in nodes with different labels being mapped too closely in latent space, making them difficult to distinguish during prompting. However, tuning GNN parameters directly during prompting presents other challenges, including computational inefficiency and the risk of over-fitting due to sparse downstream labels [19]. A theoretical analysis is offered in section 4, with experimental results in subsection E.6. A theoretical analysis of these issues is provided in section 4, with corresponding experimental results in subsection E.6.

To efficiently adapt to the distributions of downstream tasks while preserving the knowledge in the pre-trained weights, we introduce the Graph Low-Rank Adaptation (GLoRA) module, inspired by LoRA from the NLP field [11]. GLoRA targets two components during fine-tuning: (i) the message-passing scheme and (ii) the projection matrices. Formally, for the $l$-th GNN layer, the fine-tuning process with GLoRA is expressed as follows:

$$\mathbf{H}^{(l)} = \left( \mathbf{A} + \mathbf{P}_A^{(l)} \mathbf{Q}_A^{(l)\top} \right) \mathbf{H}^{(l-1)} \left( \mathbf{W}_0^{(l)} + \mathbf{P}^{(l)} \mathbf{Q}^{(l)\top} \right), \quad (5)$$

where $\mathbf{W}_0^{(l)}$ represents the frozen parameters of the $l$-th GNN layer. $\mathbf{P}^{(l)}, \mathbf{Q}^{(l)} \in \mathbb{R}^{d \times r}$ and $\mathbf{P}_A^{(l)}, \mathbf{Q}_A^{(l)} \in \mathbb{R}^{N \times 1}$ denotes the trainable low-rank adaptation matrices of layer $l$ with rank of $r$ and 1, respectively. Note that $r << d$, and for extremely large graphs, $\mathbf{P}_A^{(l)} \mathbf{Q}_A^{(l)\top}$ can be further reduced, see Appendix A for details.

The adaptation in GLoRA operates on two levels: (i) $\mathbf{P}_A$ and $\mathbf{Q}_A$ adjust the message-passing process, allowing for more effective alignment with downstream tasks by modulating the connections between nodes; and (ii) $\mathbf{P}$ and $\mathbf{Q}$ adapt the projection matrices. This dual adaptation allows DAGPrompT to handle diverse downstream task distributions, such as disentangling embeddings of connected nodes with different labels. It retains the benefits of pre-training, including efficient few-shot learning and adaptability to new tasks. Meanwhile, the frozen parameters preserve the knowledge acquired during pre-training.

### 3.2.2 Hop-specific Graph Prompting.

*Unification of Downstream Tasks.* We begin the elaboration of our prompting technique by introducing how we unify various downstream tasks. To achieve this, we reformulate all downstream tasks as **sub-graph** level tasks, as they represent a general and expressive framework for many tasks [28]. This allows us to adapt various downstream tasks to our link-prediction pre-training task. Formally, given a node $v$ in a graph $\mathcal{G}$, we define its $k$-hop neighborhood as $\mathcal{N}_k(v)$ and its embedding (produced by the GNN encoder $f$) as $\mathbf{s}_{k,v}$. Consequently, we have:

- **Link-Prediction.** Given a node triplet $(v, a, b)$ where an edge exists between nodes $(v, a)$ but not between $(v, b)$ does not, it's expected that $\text{sim}(\mathbf{s}_{k,v}, \mathbf{s}_{k,a}) > \text{sim}(\mathbf{s}_{k,v}, \mathbf{s}_{k,b})$. Here, the similarity measure (sim) can be computed using methods such as cosine similarity.
- **Node Classification.** In a graph with $C$ labels, we construct $C$ pseudo-nodes, with their embeddings initialized as the mean of the embeddings of nodes from the same class in the training set. The label prediction task for a node $v$ is then reduced to identifying the pseudo-node most likely to

form an edge with $v$, transforming the problem into a link prediction task.
- **Graph Classification.** For a set of graphs with $C$ labels, we generate $C$ pseudo-graphs, initializing their embeddings as the average of the graph embeddings from the training set. Similar to node classification, predicting a graph's label is formulated as a link prediction problem between the graph and the pseudo-graphs.

Conventional approaches with GNN encoders of $L$ layers typically rely on the final layer embedding $\mathbf{H}^{(L)}$, or a combination of all intermediate embeddings for prompting [9, 18, 27]. However, these methods often fail to account for hop-specific preferences of different nodes, limiting their adaptability. For example, in heterophilic graphs like dating networks where gender is the label, the first-hop neighborhood may exhibit heterophily, while the second-hop neighborhood may show homophily [20, 42]. We illustrate this in Figure 7. Given the varying distributions across hops and their potential differing impact on performance [42], we propose decoupling the graph prompting process in a hop-specific manner.

First, we collect intermediate embeddings from GNN layers to construct a more informative sequence than using only $\mathbf{H}^{(L)}$:

$$\mathbf{H} = \left[ \mathbf{H}^{(0)} \| \mathbf{H}^{(1)} \| \cdots \| \mathbf{H}^{(L)} \right] \in \mathbb{R}^{(L+1) \times N \times d}, \quad (6)$$

where $\mathbf{H}^{(l)}$ represents the embedding produced by the $l$-th layer of the GNN encoder, and $\mathbf{H}^{(0)} = \text{Linear}(\mathbf{X})$. Then, we gather the *Layer-specific Class Prompts* from each layer:

$$\mathbf{P} = \left[ \mathbf{P}^{(0)} \| \mathbf{P}^{(1)} \| \cdots \| \mathbf{P}^{(L)} \right] \in \mathbb{R}^{(L+1) \times C \times d}$$

$$\mathbf{P}_c^{(l)} = \frac{1}{|\mathcal{D}_c^{\text{train}}|} \sum_{v \in \mathcal{D}_c^{\text{train}}}^{|\mathcal{D}_c^{\text{train}}|} \mathbf{H}_v^{(l)} + \mathbf{\Theta}_c^{(l)}, \quad (7)$$

where $C$ denotes the number of classes in the dataset, $\mathcal{D}_c^{\text{train}}$ represents the subset of the training dataset with label $c$, and $\mathbf{\Theta}^{(l)} \in \mathbb{R}^{C \times d}$ is a layer-specific learnable prompt that enhances the model's representational capacity. These Layer-specific Class Prompts capture the hop-specific representations of the training nodes, allowing for more precise prompting and evaluation at each hop.

Based on the two sequences of node embeddings $\mathbf{H}$, and class tokens $\mathbf{P}$, we prompt the graph in a hop-specific manner, effectively addressing the diverse hop-wise distributions present in graphs:

$$\mathbf{S}^{(l)} = \text{Sim}\left( \mathbf{H}^{(l)}, \mathbf{P}^{(l)} \right), l = 0, 1, \cdots, L, \quad (8)$$

where $\text{Sim}(\cdot, \cdot)$ is a similarity function, for which we adopt cosine similarity, and $\mathbf{S}^{(l)}$ represents the prompted scores at hop $l$. Finally, we introduce a set of learnable coefficients to adaptively integrate these scores and obtain the final result:

$$\tilde{\mathbf{S}} = \sum_{l=1}^{L} \gamma^{(l)} \mathbf{S}^{(l)}, \quad \hat{\mathbf{Y}} = \arg\max_c \tilde{\mathbf{S}}_c, \quad (9)$$

where $\tilde{\mathbf{S}}_c$ denotes the score for label $c$, and $\gamma^{(l)} \in \mathbb{R}$ is a learnable parameter, initially set as $\gamma^{(l)} = \alpha(1-\alpha)^l$ with $\gamma^{(L)} = (1-\alpha)^L$, where $\alpha \in [0, 1]$ is a hyper-parameter. This setup incorporates prior knowledge about the relative importance of different hops in the graph, controlled by $\alpha$. For instance, by tuning $\alpha$, one can prioritize

closer hops over distant ones, or adjust the relative importance between them. During training, DAGPrompT adaptively refines $\gamma^{(l)}$, which helps handle the diverse distributions across hops, improving model performance. Additionally, it remains computationally efficient, requiring only a small number of additional parameters.

Finally, we adopt the following loss to optimize the parameters with a temperature $\tau$ and a cosine similarity function $\text{Sim}(\cdot, \cdot)$:

$$\mathcal{L}_{ds} = -\sum_{l=0}^{L} \sum_{(x_i, y_i) \in \mathcal{D}^{train}} \frac{\exp\left(\text{Sim}\left(\mathbf{H}_{x_i}^{(l)}, \mathbf{P}_{y_i}^{(l)}\right)/\tau\right)}{\sum_{c=1}^{C} \exp\left(\text{Sim}\left(\mathbf{H}_{x_i}^{(l)}, \mathbf{P}_{c}^{(l)}\right)/\tau\right)} \quad (10)$$

## 4 Theoretical Analysis

In this section, we present a theoretical analysis of the GLoRA module. Although low-rank adaptation may be less optimal than full-parameter fine-tuning in NLP tasks [31], we demonstrate that in few-shot settings, low-rank adaptation proves to be more effective.

THEOREM 1. *Let $\mathcal{H}$ be a hypothesis class, and $\mathcal{D} = \{(x_i, y_i)\}$ be a dataset of $m$ i.i.d. samples. Suppose the loss function $\ell(h(x), y)$ is bounded by $0 \leq \ell(h(x), y) \leq B$. Then, with probability at least $1 - \delta$, for all $h \in \mathcal{H}$, we have:*

$$L(h) - \hat{L}_{\mathcal{D}}(h) \leq 2\mathcal{R}_{\mathcal{D}}(\mathcal{H}) + 3B\sqrt{\frac{\log(2/\delta)}{2m}},$$

*where $L(h)$ is the true risk, $\hat{L}_{\mathcal{D}}(h)$ is the empirical risk, and $\mathcal{R}_{\mathcal{D}}(\mathcal{H})$ is the empirical Rademacher complexity [25].*

When data is limited, the second term grows large due to the small $m$, making it crucial to minimize the first term, which is influenced by model complexity. Low-rank adaptations like GLoRA reduce model complexity by using much fewer parameters, tightening the generalization bound, and improving performance in few-shot settings. In contrast, freezing all parameters (resulting in zero complexity) leads to high empirical risk $\hat{L}_{\mathcal{D}}(h)$ and underfitting, which is sub-optimal. GLoRA strikes a balance between flexibility and complexity, enhancing generalization in limited data scenarios. The experiment in subsection E.6 supports this analysis.

## 5 Experiments

In this section, we evaluate the capability of DAGPrompT by addressing the following key questions:

- **Q1:** How does DAGPrompT perform compared to state-of-the-art models on real-world datasets?
- **Q2:** How does the internal data distribution, such as heterophily levels, affect DAGPrompT's performance?
- **Q3:** How does the number of labels impact DAGPrompT's performance?
- **Q4:** How well does DAGPrompT transfer to other graphs?
- **Q5:** What is the running efficiency of DAGPrompT?
- **Q6:** How do the main components of DAGPrompT influence its performance?
- **Q7:** How does fine-tuning with GLoRA benefit learning?

We also conduct additional experiments on other backbones, along with a full-shot evaluation, parameter analysis, and visualizations of graph hop-wise distributions and GLoRA weights, as detailed in Appendix E.

### 5.1 Datasets and Settings

**Table 1: Statistics for node-classification datasets. $h$ stands for the homophily ratio.**

| Dataset | #Nodes | #Edges | #Attributes | #Class | $h$ |
|---------|--------|--------|-------------|--------|-----|
| Texas | 183 | 325 | 1703 | 5 | 0.11 |
| Wisconsin | 251 | 515 | 1703 | 5 | 0.20 |
| Cornell | 183 | 298 | 1703 | 5 | 0.30 |
| Chameleon | 2277 | 36101 | 1703 | 5 | 0.20 |
| Squirrel | 5201 | 217073 | 2089 | 5 | 0.22 |
| Arxiv-year | 169343 | 1166243 | 128 | 5 | 0.22 |
| Cora | 2708 | 10556 | 1433 | 7 | 0.81 |

**Table 2: Statistics for graph-classification datasets.**

| Dataset | #Graphs | #Avg.Nodes | #Avg.Edges | #Attributes | #Class |
|---------|---------|------------|------------|-------------|--------|
| Texas* | 183 | 10.5 | 9.96 | 1703 | 5 |
| Chameleon* | 2277 | 16.3 | 31.3 | 1703 | 5 |
| MUTAG | 177 | 17.9 | 19.7 | 7 | 2 |
| COX2 | 467 | 41.2 | 43.5 | 3 | 2 |
| ENZYMES | 600 | 32.6 | 62.2 | 3 | 6 |

We evaluate DAGPrompT on both few-shot node classification and graph classification tasks. For the few-shot node classification, we use seven datasets of varying scales, types, and heterophily levels. Texas, Wisconsin, Cornell, Chameleon, Squirrel [23], and Arxiv-Year [17] represent well-known heterophily datasets, while Cora [35] is a commonly used homophily graph. The dataset statistics are provided in Table 1. Additionally, we generate synthetic graphs with varying levels of heterophily following [20]. Classification accuracy is measured on five-shot and ten-shot settings for all datasets. For few-shot graph classification, we adapt the Texas and Chameleon datasets (denoted with *) by sampling each node's 2-hop neighbors and labeling each graph with the center node's label. We also include three molecular datasets—MUTAG, COX2, and ENZYMES [15]—for comparison, they are evaluated under a five-shot setting. Further details can be found in Appendix D.

### 5.2 Involved Baselines & Settings

To thoroughly evaluate the effectiveness of DAGPrompT, we compare it with several state-of-the-art baselines, categorized as follows:

- **Supervised.** We train GCN [13] from scratch, which is widely used for homophily graphs. Additionally, we include heterophily-aware GNNs such as H2GCN [42], GPR-GNN [6], and ALT-GNN [33] for comparison.
- **Pre-training + Fine-tuning.** We pre-train GCN using link prediction (LP) [19], DGI [30], and GraphCL [36], and then fine-tune the models on downstream tasks.
- **Pre-training + Prompting.** We pre-train GNNs using the link-prediction task (or the task specified by each model) and prompt them with graph prompting techniques. The graph prompting methods considered include GPPT [27], GraphPrompt [18], GPF, GPF-Plus [9], All-In-One [28], HG-Prompt [37], and GCOPE [41].

We employ the Adam Stochastic Gradient Descent optimizer [8] with a learning rate $\eta \in \{0.1, 0.5, 1, 5, 10\} \times 10^{-4}$, a weight decay in $\{0, 2.5, 5\} \times 10^{-6}$, and a maximum of 200 epochs to train all models. The dimensions of hidden representations are set to 128 or 256. We choose $r$ in $\{8, 16, 32\}$, and $\alpha$ in $\{0.1, 0.3, 0.5, 0.7, 0.9\}$. Hyper-parameters are selected based on performance. We train all the models on five NVIDIA RTX4090 with 24G memory. For a fair comparison, we use GCN as the backbone for all models except for GCOPE, where FAGCN [2] is recommended. For GPF and GPF-Plus, we choose the link-prediction task for pre-training, referring to them as GPF-LP and GPF-Plus-LP. The pre-training strategies for other methods follow the approaches outlined in their original papers.

### 5.3 Evaluation on Real-world Datasets (Q1)

We evaluate DAGPrompT with GCN as the backbone, as shown in Table 3, and draw the following key observations: (i) **DAGPrompT consistently outperforms other baselines by a large margin.** The performance improvements on heterophily datasets are particularly notable, with up to a 7.55% increase on Texas and an average improvement of 3.63% across all datasets. (ii) **Fine-tuning or prompting methods sometimes underperform compared to training from scratch on heterophily graphs.** For example, H2GCN, trained from scratch, surpasses most graph prompt methods on Texas, Cornell, and Wisconsin. This supports the claim in section 1 that larger gaps exist between pre-training and downstream tasks in complex graphs, where task reformulation and prompting alone are insufficient to bridge the gap caused by intricate graph distributions. The heterophily in these graphs limits the effectiveness of prompt-based methods in learning embeddings. (iii) **Fewer labels for training significantly hinder models trained from scratch, especially in heterophily settings.** On graphs with smaller label ratios, Chameleon, Squirrel, and Arxiv-year, even non-heterophily-aware models, such as GraphPrompt, outperform heterophily-aware models by a large margin. This may be due to better utilization of graph structure during the pre-training and fine-tuning phases.

We also conduct experiments on the graph classification task[1], as shown in Table 4, where DAGPrompT consistently delivers the best performance.

### 5.4 Evaluation on Data Heterophily (Q2) and Number of Shots (Q3)

We investigate the impact of varying heterophily levels by generating a series of synthetic graphs, Syn-Chameleon, based on the Chameleon dataset. Following the method in [20], we control the homophily ratio of Syn-Chameleon by adjusting the edges, allowing the homophily ratio to range from 0.9 (strong homophily) to 0.1 (strong heterophily). Models are evaluated on these graphs under a *full-shot* setting, using 50% of the nodes for training to minimize the effect of label quantity. The results, shown in Figure 4, demonstrate that DAGPrompT consistently outperforms the baselines, especially in strong heterophily scenarios. Notably, heterophily-aware models like GPR-GNN outperform most non-heterophily-aware models, such as GraphPrompt and GPF-Plus-LP, in this setting.

---

[1]Baselines unsuitable for graph classification are excluded.

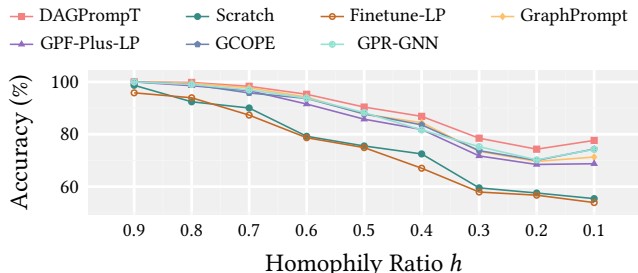

**Figure 4: Impact of data heterophily on Syn-Chameleon.**

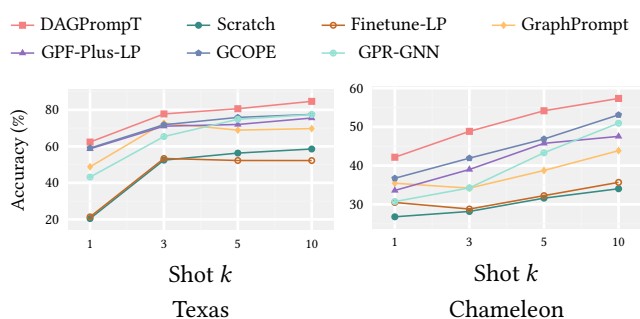

**Figure 5: Impact of shots on Texas and Chameleon.**

We evaluate the effect of varying shot numbers on the Texas and Chameleon datasets, as shown in Figure 5, by adjusting the shot count from 1 to 10. Overall, DAGPrompT consistently outperforms the baselines, with the most notable improvements occurring at 5-shot on Texas (5.51%) and 3-shot on Chameleon (3.99%). As the number of shots increases, the non-pre-trained model, GPR-GNN, demonstrates a considerable advantage over prompt-based methods. However, in scenarios with extremely limited labeled data, GPR-GNN underperforms significantly, lagging behind prompting approaches.

These two experiments reinforce the findings from subsection 5.3: heterophily and label scarcity are key factors that limit model performance. Prompting methods address label scarcity but overlook heterophily, while heterophily-oriented methods generally neglect label scarcity. If either issue is inadequately addressed, performance declines significantly, underscoring the need for a distribution-aware graph prompting approach.

### 5.5 Evaluation on Transfer Ability (Q4)

We evaluate the transferability of DAGPrompT in Table 5. For pre-training, we use the Texas dataset as the source domain and test the transfer to downstream tasks on the Cornell, Wisconsin, and Chameleon datasets. Models with the suffix *-Scratch* are those trained directly on the downstream tasks without pre-training, while models with the suffix *-Cross* are pre-trained on the source domain and then fine-tuned (or prompted) on the target domains.

The results show that pre-training, even across different domains, generally enhances model performance. DAGPrompT exhibits the most significant improvement when transitioning from training

**Table 3: Summary of the mean and standard deviation of accuracy across all runs for the few-shot node classification tasks. The best results for each dataset are highlighted in gray. GCN is used as the backbone encoder, except for H2GCN, GPR-GNN, ALT-GNN, and GCOPE, which use their specific architectures.**

| | Texas (0.11) | | Cornell (0.31) | | Wisconsin (0.20) | | Chameleon (0.20) | | Squirrel (0.22) | | Arxiv-year (0.22) | | Cora (0.81) | |
|---|---|---|---|---|---|---|---|---|---|---|---|---|---|---|
| | 5-shot | 10-shot | 5-shot | 10-shot | 5-shot | 10-shot | 5-shot | 10-shot | 5-shot | 10-shot | 5-shot | 10-shot | 5-shot | 10-shot |
| Label Ratio | 13.6% | 27.3% | 13.6% | 27.3% | 9.9% | 19.9% | 1.09% | 2.19% | 0.48% | 0.96% | 0.0147% | 0.0295% | 1.29% | 2.58% |
| GCN | $56.32_{\pm3.45}$ | $58.55_{\pm4.92}$ | $59.01_{\pm4.10}$ | $63.42_{\pm3.24}$ | $56.24_{\pm3.97}$ | $56.37_{\pm3.12}$ | $31.61_{\pm3.32}$ | $34.04_{\pm4.42}$ | $21.07_{\pm0.82}$ | $23.01_{\pm1.73}$ | $22.05_{\pm1.36}$ | $23.94_{\pm1.34}$ | $55.27_{\pm1.34}$ | $58.45_{\pm1.74}$ |
| H2GCN | $72.95_{\pm3.30}$ | $76.75_{\pm4.74}$ | $77.86_{\pm4.82}$ | $82.62_{\pm5.86}$ | $68.38_{\pm2.95}$ | $72.72_{\pm3.74}$ | $44.74_{\pm2.95}$ | $49.64_{\pm2.75}$ | $29.47_{\pm2.43}$ | $30.61_{\pm3.48}$ | $28.65_{\pm1.44}$ | $30.23_{\pm1.37}$ | $66.56_{\pm1.37}$ | $69.45_{\pm1.42}$ |
| GPR-GNN | $74.83_{\pm2.84}$ | $77.44_{\pm3.96}$ | $79.73_{\pm3.85}$ | $82.87_{\pm4.02}$ | $71.42_{\pm3.28}$ | $73.24_{\pm2.58}$ | $43.32_{\pm2.52}$ | $50.96_{\pm3.28}$ | $28.93_{\pm1.36}$ | $32.05_{\pm2.86}$ | $28.49_{\pm2.94}$ | $30.47_{\pm2.41}$ | $69.03_{\pm1.40}$ | $71.42_{\pm1.08}$ |
| ALT-GNN | $74.36_{\pm2.86}$ | $78.42_{\pm2.76}$ | $78.72_{\pm3.46}$ | $83.51_{\pm3.75}$ | $70.45_{\pm2.38}$ | $73.52_{\pm2.01}$ | $45.51_{\pm2.47}$ | $52.84_{\pm1.98}$ | $30.73_{\pm2.23}$ | $33.26_{\pm2.97}$ | $29.05_{\pm2.74}$ | $30.20_{\pm1.98}$ | $68.15_{\pm1.09}$ | $71.30_{\pm1.42}$ |
| Finetune-LP | $52.26_{\pm4.42}$ | $52.23_{\pm6.15}$ | $56.20_{\pm4.66}$ | $61.53_{\pm5.57}$ | $55.27_{\pm3.85}$ | $57.84_{\pm2.90}$ | $32.26_{\pm4.21}$ | $35.69_{\pm4.28}$ | $22.55_{\pm1.52}$ | $24.29_{\pm2.38}$ | $22.26_{\pm2.16}$ | $24.29_{\pm1.51}$ | $60.31_{\pm2.06}$ | $65.26_{\pm2.07}$ |
| Finetune-DGI | $50.32_{\pm5.84}$ | $51.52_{\pm9.17}$ | $54.36_{\pm6.31}$ | $62.61_{\pm6.08}$ | $51.45_{\pm5.66}$ | $54.95_{\pm6.02}$ | $32.55_{\pm1.92}$ | $36.58_{\pm3.07}$ | $22.27_{\pm1.87}$ | $24.41_{\pm1.19}$ | $23.22_{\pm1.96}$ | $24.80_{\pm1.72}$ | $61.78_{\pm2.32}$ | $64.75_{\pm3.97}$ |
| Finetune-GCL | $46.17_{\pm4.13}$ | $49.54_{\pm6.89}$ | $52.06_{\pm4.81}$ | $61.98_{\pm3.10}$ | $46.39_{\pm5.68}$ | $52.48_{\pm9.52}$ | $30.13_{\pm2.42}$ | $36.85_{\pm5.37}$ | $21.07_{\pm5.86}$ | $22.46_{\pm4.97}$ | $22.19_{\pm3.48}$ | $22.91_{\pm7.01}$ | $53.91_{\pm2.96}$ | $57.81_{\pm2.85}$ |
| GPPT | $54.56_{\pm5.24}$ | $59.93_{\pm4.37}$ | $56.79_{\pm6.02}$ | $62.47_{\pm5.37}$ | $53.57_{\pm2.48}$ | $53.94_{\pm3.21}$ | $38.75_{\pm1.55}$ | $43.86_{\pm2.92}$ | $25.78_{\pm2.23}$ | $28.32_{\pm2.86}$ | $24.45_{\pm1.19}$ | $25.08_{\pm1.47}$ | $63.34_{\pm1.84}$ | $65.96_{\pm1.32}$ |
| GraphPrompt | $68.90_{\pm1.95}$ | $69.73_{\pm2.02}$ | $72.38_{\pm6.89}$ | $79.62_{\pm6.99}$ | $66.88_{\pm2.25}$ | $68.09_{\pm2.76}$ | $47.89_{\pm4.17}$ | $52.84_{\pm2.77}$ | $30.23_{\pm3.87}$ | $32.93_{\pm2.45}$ | $28.46_{\pm1.02}$ | $28.76_{\pm2.18}$ | $70.21_{\pm1.35}$ | $71.74_{\pm0.97}$ |
| GPF-LP | $66.93_{\pm6.06}$ | $66.01_{\pm0.82}$ | $69.14_{\pm7.40}$ | $72.01_{\pm8.35}$ | $59.46_{\pm2.59}$ | $61.67_{\pm2.43}$ | $45.76_{\pm2.16}$ | $47.55_{\pm3.03}$ | $28.57_{\pm2.79}$ | $29.45_{\pm1.90}$ | $28.64_{\pm5.82}$ | $29.03_{\pm3.84}$ | $65.30_{\pm2.45}$ | $67.31_{\pm2.94}$ |
| GPF-Plus-LP | $71.99_{\pm4.41}$ | $75.51_{\pm2.38}$ | $78.17_{\pm8.48}$ | $82.24_{\pm3.97}$ | $68.26_{\pm4.32}$ | $72.82_{\pm2.45}$ | $49.40_{\pm3.21}$ | $53.37_{\pm2.89}$ | $31.08_{\pm2.06}$ | $33.29_{\pm2.45}$ | $29.45_{\pm3.32}$ | $30.06_{\pm1.72}$ | $69.43_{\pm1.09}$ | $70.85_{\pm1.86}$ |
| All-In-One | $71.85_{\pm3.08}$ | $74.70_{\pm2.37}$ | $79.42_{\pm5.27}$ | $81.37_{\pm4.72}$ | $69.63_{\pm3.09}$ | $70.18_{\pm2.67}$ | $48.09_{\pm2.97}$ | $53.63_{\pm2.84}$ | $30.22_{\pm2.01}$ | $32.58_{\pm2.74}$ | $29.85_{\pm3.62}$ | $30.89_{\pm2.84}$ | $67.04_{\pm2.01}$ | $70.42_{\pm1.64}$ |
| HGPrompt | $67.48_{\pm2.08}$ | $70.30_{\pm2.02}$ | $72.01_{\pm5.33}$ | $73.47_{\pm6.38}$ | $65.87_{\pm3.27}$ | $66.08_{\pm3.58}$ | $46.87_{\pm3.29}$ | $53.10_{\pm3.04}$ | $30.09_{\pm2.33}$ | $32.46_{\pm2.98}$ | $28.41_{\pm1.34}$ | $28.90_{\pm2.08}$ | $68.42_{\pm1.37}$ | $69.54_{\pm1.54}$ |
| GCOPE | $75.85_{\pm2.36}$ | $77.50_{\pm1.94}$ | $78.53_{\pm4.74}$ | $82.04_{\pm5.36}$ | $71.45_{\pm2.86}$ | $73.85_{\pm2.84}$ | $49.24_{\pm3.37}$ | $54.01_{\pm2.74}$ | $31.32_{\pm2.45}$ | $34.06_{\pm2.45}$ | $29.59_{\pm1.45}$ | $30.67_{\pm1.98}$ | $69.24_{\pm1.35}$ | $70.57_{\pm2.64}$ |
| DAGPrompT | $81.36_{\pm4.93}$ | $83.10_{\pm2.92}$ | $87.28_{\pm1.64}$ | $89.30_{\pm1.44}$ | $76.37_{\pm1.17}$ | $76.52_{\pm2.84}$ | $53.38_{\pm1.97}$ | $58.29_{\pm2.12}$ | $34.95_{\pm2.77}$ | $36.54_{\pm2.77}$ | $31.06_{\pm1.03}$ | $31.99_{\pm0.98}$ | $71.60_{\pm1.77}$ | $73.42_{\pm0.43}$ |
| *Improvement* | 5.51 | 4.68 | 7.55 | 5.79 | 4.92 | 2.67 | 3.98 | 4.28 | 3.63 | 2.48 | 1.21 | 1.10 | 1.39 | 1.68 |

**Table 4: Summary of the mean and standard deviation of accuracy across all runs for the graph classification. The best results for each dataset are highlighted in gray.**

| | Texas* | Chameleon* | MUTAG | COX2 | ENZYMES |
|---|---|---|---|---|---|
| Scratch | $52.46_{\pm2.34}$ | $25.30_{\pm2.84}$ | $56.43_{\pm2.85}$ | $45.98_{\pm4.97}$ | $22.65_{\pm3.85}$ |
| Finetune-LP | $53.37_{\pm1.84}$ | $25.75_{\pm2.85}$ | $58.87_{\pm1.65}$ | $51.45_{\pm3.65}$ | $24.90_{\pm4.74}$ |
| GraphPrompt | $72.75_{\pm2.09}$ | $44.18_{\pm2.51}$ | $73.85_{\pm1.97}$ | $55.86_{\pm5.73}$ | $25.67_{\pm3.49}$ |
| GPF-LP | $68.34_{\pm2.14}$ | $41.01_{\pm3.30}$ | $70.68_{\pm2.75}$ | $40.87_{\pm5.67}$ | $20.58_{\pm1.97}$ |
| GPF-Plus-LP | $71.06_{\pm2.56}$ | $46.27_{\pm4.41}$ | $73.86_{\pm1.90}$ | $54.80_{\pm3.48}$ | $25.65_{\pm3.97}$ |
| All-In-One | $73.46_{\pm1.90}$ | $46.26_{\pm2.85}$ | $74.58_{\pm1.85}$ | $55.03_{\pm3.48}$ | $26.08_{\pm4.86}$ |
| HGPrompt | $70.56_{\pm2.86}$ | $45.93_{\pm2.38}$ | $73.46_{\pm1.37}$ | $50.07_{\pm4.87}$ | $22.75_{\pm4.87}$ |
| GCOPE | $73.64_{\pm2.11}$ | $46.78_{\pm2.85}$ | $73.98_{\pm2.64}$ | $52.18_{\pm3.38}$ | $25.45_{\pm5.38}$ |
| DAGPrompT | $79.53_{\pm2.89}$ | $51.26_{\pm3.44}$ | $76.01_{\pm1.79}$ | $56.46_{\pm4.76}$ | $26.71_{\pm4.22}$ |
| *Improvement* | 5.89 | 4.48 | 1.43 | 0.60 | 0.63 |

**Table 5: Transfer ability measured by classification accuracy across different domains. Source domain: Texas. Target domains: Cornell, Wisconsin, and Chameleon.**

| | Cornell | Wisconsin | Chameleon |
|---|---|---|---|
| Finetune-LP-Scratch | $59.01_{\pm4.10}$ | $56.24_{\pm3.97}$ | $31.61_{\pm3.32}$ |
| Finetune-LP-Cross | $60.50_{\pm2.30}$ | $56.58_{\pm1.90}$ | $31.18_{\pm2.81}$ |
| All-In-One-Scratch | $64.16_{\pm2.13}$ | $63.96_{\pm2.90}$ | $32.56_{\pm2.13}$ |
| All-In-One-Cross | $75.57_{\pm2.47}$ | $66.85_{\pm2.48}$ | $45.86_{\pm2.41}$ |
| DAGPrompT-Scratch | $65.58_{\pm1.65}$ | $63.34_{\pm2.95}$ | $33.45_{\pm2.09}$ |
| DAGPrompT-Cross | $82.45_{\pm2.74}$ | $70.75_{\pm2.84}$ | $52.08_{\pm1.38}$ |

from scratch to cross-domain pre-training, highlighting its strong transferability. This makes DAGPrompT particularly useful in cases where initial training data is unavailable or unsuitable due to privacy concerns or computational constraints.

## 5.6 Efficiency Analysis (Q5)

**Table 6: Efficiency analysis on the Chameleon dataset, reporting iterations per second, peak GPU memory (MB), and the number of tunable parameters. The average rank across these metrics is also provided. "PT" refers to pre-training, "DS" to downstream, "-" indicates not applicable, and "K" represents thousand.**

| | Iter/sec. ↑ | | Memory ↓ | | T.Parameters ↓ | | Avg.Rank ↓ |
|---|---|---|---|---|---|---|---|
| | PT | DS | PT | DS | PT | DS | |
| Scratch | - | 72.82 | - | 889 | - | 331K | #3.3 |
| Finetune-LP | 2.82 | 69.68 | 586 | 896 | 331K | 331K | #3.3 |
| GPPT | 1.30 | 5.35 | 583 | 1527 | 331K | 3.8K | #4.2 |
| GraphPrompt | 2.57 | 37.81 | 591 | 2101 | 331K | 2K | #3.2 |
| GPF-Plus-LP | 2.09 | 34.97 | 591 | 2515 | 331K | 93.5K | #5.2 |
| All-In-One | 1.86 | 20.45 | 853 | 3064 | 331K | 7.4K | #6.3 |
| HGPrompt | 2.49 | 24.57 | 585 | 2908 | 331K | 2K | #4.0 |
| GCOPE | 0.49 | 1.37 | 525 | 2184 | 92.5K | 47.6K | #5.2 |
| DAGPrompT | 2.60 | 27.54 | 585 | 2121 | 331K | 6.4K | #3.5 |

We evaluate the efficiency of DAGPrompT on the Chameleon dataset, as shown in Table 6. The results demonstrate that DAGPrompT is generally efficient, exhibiting fast running speed, low GPU memory consumption, and a small number of tunable parameters. Overall, DAGPrompT ranks highly in terms of time efficiency, memory usage, and parameter efficiency.

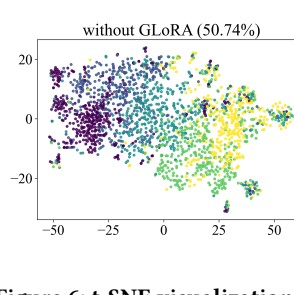 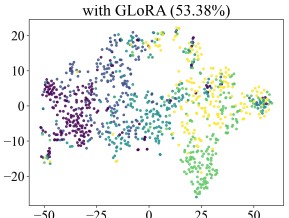

**Figure 6: t-SNE visualization of GNN encoder embeddings on the Chameleon dataset, with and without GLoRA.**

## 5.7 Ablation Study (Q6)

**Table 7: Ablation Study on Texas, Cornell, and Chameleon. "w/o" denotes without.**

|  | Texas | Cornell | Chameleon |
|---|---|---|---|
| DAGPrompT | $81.36_{\pm 4.93}$ | $87.28_{\pm 3.63}$ | $53.38_{\pm 1.97}$ |
| w/o GLoRA | $79.01_{\pm 4.29}$ | $84.44_{\pm 2.04}$ | $50.74_{\pm 1.43}$ |
| w/o Layer-Specific Prompts | $79.46_{\pm 3.95}$ | $84.42_{\pm 3.46}$ | $53.17_{\pm 2.47}$ |
| w/o Coefficients $\gamma$ | $78.45_{\pm 3.84}$ | $82.08_{\pm 3.94}$ | $51.47_{\pm 2.47}$ |

We conduct an ablation study to evaluate the contribution of each component in DAGPrompT, as detailed in Table 7, by disabling them individually. For the coefficients $\gamma$ in Equation 9, we fix all values to 1. The results show that GLoRA significantly boosts performance. Additionally, the coefficients $\gamma$ further enhance results. Overall, each component contributes to performance to varying extents.

## 5.8 GLoRA Visualizations (Q7)

We extract embeddings from the GNN encoder on the Chameleon dataset, both with and without GLoRA, and visualize them using t-SNE, as shown in Figure 6. The results show that GLoRA enhances the GNN encoder's ability to adapt to new distributions during fine-tuning and prompting. With GLoRA, the label clusters are more compact and separable, indicating that the embeddings are adjusted according to label information in heterophily graphs. The issue illustrated in Figure 1, where connected nodes with different labels have indistinguishable embeddings, is mitigated. This improved separability facilitates classification, contributing to a 2.64% performance increase compared to the variant without GLoRA.

## 6 Related Works

### 6.1 Graph Pre-training and Prompting

In recent years, significant advancements have been made in the development of pre-trained Graph Neural Networks (GNNs). These methods can be broadly categorized into three main types: (i) *Graph Property Reconstruction-Based Methods*, which focus on reconstructing specific graph properties such as node attributes [10, 12] or links [14, 19]; (ii) *Sub-Graph Contrastive Methods*, which distinguish positive subgraphs from negative ones [36, 40, 44]; and (iii) *Local-Global Contrastive Methods*, which leverage mutual information to encode global patterns in local representations [26, 30].

The aforementioned approaches often overlook the objective gap between pre-training and fine-tuning, which limits their generalization across tasks [28]. To address this, recent studies have adopted prompting techniques inspired by advances in Natural Language Processing fields [3, 4]. GPPT [27] was the first to incorporate learnable graph label prompts, reformulating downstream node classification tasks as link prediction tasks to narrow the gap between pre-training and fine-tuning. VNT [29] introduces prompts specifically tailored for pre-trained graph transformers in node classification tasks. Subsequently, GraphPrompt [18] introduced a unified template to accommodate a broader range of downstream tasks. All-in-One [28] reformulated all downstream tasks into graph-level tasks and integrated meta-learning techniques for multi-task prompting. GPF and GPF-Plus [9] proposed a universal prompting system operating solely within the node feature space. While prior work primarily focused on downstream tasks, GCOPE [41] shifted the emphasis to the pre-training phase, combining disparate graph datasets to distill and transfer knowledge to target tasks. Additionally, HGraphPrompt [37] and HetGPT [21] extended prompting techniques to heterogeneous graph learning, broadening the scope of their application. Recent studies have also explored cross-domain prompting [34, 39, 43] and multi-task prompting[38]. However, these methods often neglect the complex distributions in graph data, resulting in performance degradation.

### 6.2 Heterophily Graph Learning

Traditional GNNs typically assume homophily (similarity between connected nodes) [22] and are less effective in heterophily graphs, where connected nodes differ significantly [42]. To address this, models such as H2GCN [42] and GPR-GNN [6] enhance message-passing with high-order re-weighting techniques to improve compatibility with heterophily. LINKX, a simpler model, is optimized for large-scale heterophily learning [17]. Other approaches, including GloGNN [16], GCNII [5], MWGNN [20], ALT-GNN [33], and AGS-GNN [7] refine graph convolution for heterophilous data. However, most heterophily-aware models are designed for training from scratch in label-rich scenarios and face generalization and over-fitting issues in few-shot settings.

## 7 Conclusion

In this paper, we push the limits of the graph prompting paradigm to graphs with complex distributions, such as heterophily graphs. We observe that current methods struggle to generalize in these settings and are, in some cases, outperformed by simple models trained from scratch. We identify two key challenges for better generalization on complex graphs: (i) adapting the model to new distributions in downstream tasks to reduce discrepancies between pre-training and fine-tuning due to heterophily, and (ii) aligning model prompts to the hop-specific needs of different nodes. To address these challenges, we propose Distribution-aware Graph Prompt Tuning (DAGPrompT), which includes a GLoRA module and a Hop-specific Graph Prompting module, corresponding to the two challenges outlined above. Our experiments across 10 datasets and 14 baselines demonstrate the state-of-the-art performance of DAGPrompT, achieving up to a 7.55% improvement in accuracy.

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

# A  Algorithm of DAGPrompT

We detail the DAGPrompT algorithm from pre-training to prompting in Algorithms 1 and 2. For clarity, we present the algorithm using a for-loop structure, though in practice, we process data in batches. The example provided focuses on node classification, with graph classification requiring only a straightforward adjustment: feeding entire graphs instead of sampling node neighborhoods.

---

**Algorithm 1** DAGPrompT stage one: pre-training.

---

**Input:** Node attributes $\mathbf{X}$, adjacency $\mathbf{A}$, GNN $f$ with parameter $\Theta$, temperature $\tau$, learning rate $\eta$.

**Output:** Tuned parameter $\Theta^*$.

1: **for** $i \leftarrow 1$ to $N$ **do** ▷ $N$ is the number of nodes in the graph.
2:    // Construct positive and negative samples.
3:    $v \leftarrow i$;   $a \leftarrow j \mid j \in \mathcal{N}(i)$;    $b \leftarrow k \mid k \notin \mathcal{N}(i)$;
4:    // Generate embeddings
5:    $\mathbf{H} \leftarrow f(\mathbf{X}, \mathbf{A}; \Theta)$
6:    $\mathbf{H} \leftarrow \texttt{matmul}(\mathbf{H}, \mathbf{A})$
7:    $\mathbf{s}_v \leftarrow \mathbf{H}[i]$;    $\mathbf{s}_a \leftarrow \mathbf{H}[a]$;    $\mathbf{s}_b \leftarrow \mathbf{H}[b]$
8: **end for**
9: // Loss calculation and parameter optimization
10: $\mathcal{L}_{\text{pt}}(\Theta) \leftarrow -\sum_{(v,a,b)} \ln \frac{\exp(\text{sim}(\mathbf{s}_v, \mathbf{s}_a)/\tau)}{\sum_{u \in \{a,b\}} \exp(\text{sim}(\mathbf{s}_v, \mathbf{s}_u)/\tau)}$
11: $\Theta \leftarrow \Theta - \eta \nabla_\Theta(\mathcal{L}_{\text{pt}}(\Theta))$

---

For extremely large graphs where $N \gg rd$, we enhance efficiency of GLoRA by reducing $\mathbf{P}_A^{(l)} \mathbf{Q}_A^{(l)\top}$ to a unified edge-weight vector. We then apply this edge-weight only to edges connected to nodes in the training set. This approach reduces the total number of parameters while maximizing the use of information from the training data.

## B Complexity Analysis

Consider a graph with $N$ nodes and $E$ edges, where $d$ is the hidden dimension, $L$ the number of GNN encoder layers, and $C$ the number of classes.

The pre-training complexity of the GNN encoder is $O((LE + NK)d)$, where $K$ is the number of negative samples. In this work, we set $K = 1$.

For prompting, the complexity of DAGPrompT arises from three components: (i) generating layer-wise embeddings from the GNN encoder $f$, (ii) generating layer-wise class tokens, and (iii) performing similarity calculations. Step (i) incurs a complexity of $O(L(E + Nd^2))$, driven by message passing and embedding projection. Step (ii) has a lighter complexity of $O(LCd)$, involving matrix addition. Step (iii) incurs a complexity of $O(LNCd)$, dominated by similarity calculations.

The overall complexity of DAGPrompT is $O(L(NC+LC+Nd)d + LE)$. Given that $L \ll N$ and $C \ll d$, this simplifies to $O(LNd^2 + LEd)$, yielding near-linear complexity with respect to graph size, making it efficient for large-scale applications.

## C Details of Theorem 1

Let $\mathcal{H}$ be a hypothesis class, and let $\mathcal{D} = \{(x_1, y_1), \ldots, (x_m, y_m)\}$ represent a dataset of $m$ independent and identically distributed (i.i.d.) samples drawn from an unknown distribution. The goal is to evaluate the performance of a hypothesis $h \in \mathcal{H}$, which we do by assessing its true risk (or expected error). The true risk is defined as:

$$L(h) = \mathbb{E}_{(x,y) \sim \mathcal{D}}[\ell(h(x), y)], \tag{11}$$

where $\ell(h(x), y)$ denotes a bounded loss function, satisfying $0 \leq \ell(h(x), y) \leq B$. This measures the expected loss of the hypothesis over the distribution of the data.

---

**Algorithm 2** DAGPrompT stage two: prompting and tuning, taking the node classification as an example.

---

**Input:** Node attributes $\mathbf{X}$, adjacency $\mathbf{A}$, training set $\mathcal{D}^{\text{train}}$, pre-trained GNN $f$ with parameter $\Theta^*$, GLoRA parameter $\Theta_{\text{glora}}$, layer-specific prompts $\{\Theta^{(l)}\}_{l=0}^L$, coefficients $\{\gamma^{(l)}\}_{l=0}^L$, temperature $\tau$, learning Rate $\eta$.

**Output:** downstream labels $\hat{\mathbf{Y}}$.

1: // Construct the node tokens.
2: **for** $i \leftarrow 1$ to $N$ **do** ▷ $N$ is the number of nodes in the graph.
3:    $\mathbf{X}_i, \mathbf{A}_i \leftarrow \texttt{SampleNeighborhood}(\mathbf{X}, \mathbf{A}, i)$
4:    **for** $l \leftarrow 0$ to $L$ **do**
5:      $\mathbf{H}_i^{(l)} \leftarrow f(\mathbf{X}_i, \mathbf{A}_i; \Theta^*, \Theta_{\text{glora}}; l)$ ▷ embeddings of layer $l$
6:    **end for**
7: **end for**
8: **for** $l \leftarrow 0$ to $L$ **do**
9:    $\mathbf{H}^{(l)} \leftarrow \left[ \mathbf{H}_1^{(l)} \| \mathbf{H}_2^{(l)} \| \cdots \| \mathbf{H}_N^{(l)} \right] \in \mathbb{R}^{N \times d}$
10: **end for**
11:
12: // Construct the class tokens, only calculated once.
13: **for** $c \leftarrow 1$ to $C$ **do** ▷ $C$ is the number of classes in the graph.
14:    **for** $l \leftarrow 0$ to $L$ **do**
15:      $\mathbf{P}_c^{(l)} \leftarrow \frac{1}{|\mathcal{D}_c^{\text{train}}|} \sum_{v \in \mathcal{D}_c^{\text{train}}}^{|\mathcal{D}_c^{\text{train}}|} \mathbf{H}_v^{(l)} + \Theta_c^{(l)} \in \mathbb{R}^d$
16:    **end for**
17: **end for**
18: **for** $l \leftarrow 0$ to $L$ **do**
19:    $\mathbf{P}^{(l)} \leftarrow \left[ \mathbf{P}_1^{(l)} \| \mathbf{P}_2^{(l)} \| \cdots \| \mathbf{P}_C^{(l)} \right] \in \mathbb{R}^{C \times d}$
20: **end for**
21:
22: // Prompting
23: **for** $l \leftarrow 0$ to $L$ **do**
24:    $\mathbf{S}^{(l)} \leftarrow \text{Similarity}\left( \mathbf{H}^{(l)}, \mathbf{P}^{(l)} \right)$
25: **end for**
26: $\tilde{\mathbf{S}} \leftarrow \sum_{l=1}^L \gamma^{(l)} \mathbf{S}^{(l)}$
27: $\hat{\mathbf{Y}} \leftarrow \arg\max_c \tilde{\mathbf{S}}_c$
28:
29: // Calculate loss and optimize parameters.
30: $\mathcal{L}_{\text{ds}} = -\sum_{l=0}^L \sum_{(x_i, y_i) \in \mathcal{D}^{\text{train}}} \frac{\exp\left(\text{Sim}\left(\mathbf{H}_{x_i}^{(l)}, \mathbf{P}_{y_i}^{(l)}\right)/\tau\right)}{\sum_{c=1}^C \exp\left(\text{Sim}\left(\mathbf{H}_{x_i}^{(l)}, \mathbf{P}_c^{(l)}\right)/\tau\right)}$
31: $\Theta_{\text{glora}} \leftarrow \Theta_{\text{glora}} - \eta \nabla_{\Theta_{\text{glora}}}(\mathcal{L}_{\text{ds}}(\Theta_{\text{glora}}))$
32: **for** $l \leftarrow 0$ to $L$ **do**
33:    $\Theta^{(l)} \leftarrow \Theta^{(l)} - \eta \nabla_{\Theta^{(l)}}(\mathcal{L}_{\text{ds}}(\Theta^{(l)}))$
34:    $\gamma^{(l)} \leftarrow \gamma^{(l)} - \eta \nabla_{\gamma^{(l)}}(\mathcal{L}_{\text{ds}}(\gamma^{(l)}))$
35: **end for**

---

In practice, however, we do not have access to the true distribution. Instead, we rely on the available sample $\mathcal{D}$ to estimate the performance of $h$ through the empirical risk (or training error), which is given by:

$$\hat{L}_{\mathcal{D}}(h) = \frac{1}{m} \sum_{i=1}^m \ell(h(x_i), y_i). \tag{12}$$

This empirical risk approximates the true risk by averaging the loss over the observed data points.

To understand the capacity of the hypothesis class $\mathcal{H}$ in fitting the data, we consider its empirical Rademacher complexity, which quantifies how well $\mathcal{H}$ can adapt to random noise. The empirical Rademacher complexity is defined as:

$$\mathcal{R}_{\mathcal{D}}(\mathcal{H}) = \mathbb{E}_{\sigma}\left[\sup_{h \in \mathcal{H}} \frac{1}{m} \sum_{i=1}^{m} \sigma_i h(x_i)\right], \quad (13)$$

where $\sigma_i$ are i.i.d. Rademacher variables, each taking values in $\pm 1$ with equal probability. This measure helps us understand the richness of the hypothesis class by evaluating its ability to fit random labels on the sample $\mathcal{D}$, providing insights into potential overfitting and generalization behavior.

The generalization bound for the hypothesis class $\mathcal{H}$ [25] can be formulated as:

$$L(h) \leq \hat{L}_{\mathcal{D}}(h) + 2\mathcal{R}_{\mathcal{D}}(\mathcal{H}) + 3B\sqrt{\frac{\log(2/\delta)}{2m}}, \quad (14)$$

which holds with probability at least $1 - \delta$. Here, the terms are defined as follows:

- $L(h)$: the true risk (expected test error), which reflects the hypothesis' error on unseen data,
- $\hat{L}_{\mathcal{D}}(h)$: the empirical risk (training error), representing the observed performance on the given dataset,
- $\mathcal{R}_{\mathcal{D}}(\mathcal{H})$: the empirical Rademacher complexity of the hypothesis class $\mathcal{H}$, capturing the class's capacity to fit random noise,
- $B$: the upper bound on the loss function $\ell(h(x), y)$, ensuring the loss is bounded within $[0, B]$,
- $\delta$: the confidence level, determining the probability that the bound holds,
- $m$: the number of training samples in the dataset $\mathcal{D}$.

This bound shows that the true risk $L(h)$ is upper-bounded by the empirical risk $\hat{L}_{\mathcal{D}}(h)$, adjusted by the model's complexity (as captured by the empirical Rademacher complexity $\mathcal{R}_{\mathcal{D}}(\mathcal{H})$) and a term that decreases with the number of training samples, $m$, providing insight into how well the model generalizes to unseen data.

## D  Dataset Details

In this section, we describe the datasets used in our study.

The Cora dataset [35] is a widely used citation network characterized by strong homophily [22]. In Cora, nodes represent papers, node features are bag-of-words representations derived from the content, and edges correspond to citation links. The labels indicate the subject categories of the papers.

The Texas, Wisconsin, Cornell, Chameleon, and Squirrel datasets [23] consist of web pages, where nodes represent individual pages, node features are word embeddings, and edges reflect hyperlinks. Labels for Texas, Cornell, and Wisconsin represent web page categories, while Chameleon and Squirrel labels capture average monthly web traffic, grouped into five ranges. Notably, Chameleon and Squirrel are complex Wikipedia networks, exhibiting a mix of homophily and heterophily [20].

The ArXiv-year dataset [17] is a large-scale citation network with high heterophily. Nodes represent research papers, node features are embeddings from paper titles and abstracts, and edges represent

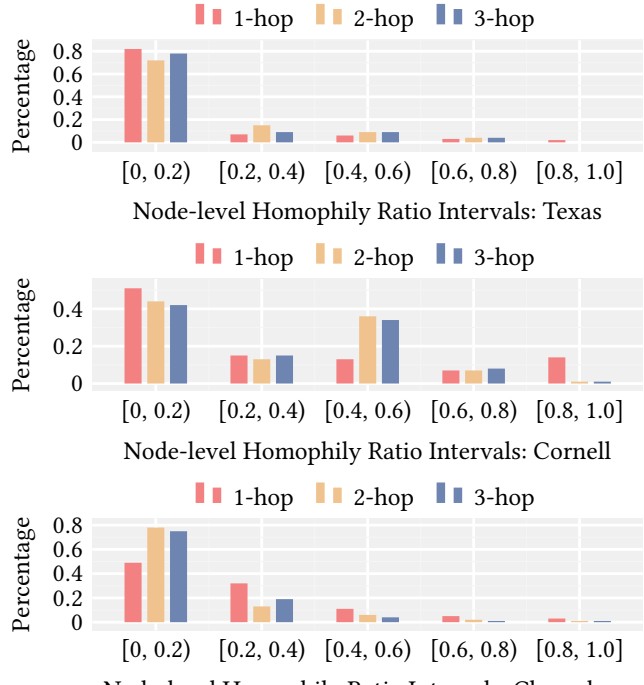

**Figure 7: Hop-wise Local Heterophily Distributions.**

citation relationships. The labels correspond to publication years, grouped into five intervals.

The MUTAG dataset consists of 188 chemical compounds, categorized into two classes based on mutagenic effects on bacteria. Here, vertices represent atoms, and edges represent chemical bonds. The COX2 dataset contains molecular structures of 467 cyclooxygenase-2 inhibitors, and the ENZYME dataset focuses on classifying 600 enzymes into six top-level EC classes [15].

We provide a statistical analysis of hop-wise distributional variance from the perspective of **local** homophily, adapted from the homophily measurement in section 2:

$$h_i^{(k)} = \frac{|\{(v_i, v_j) : (v_i, v_j) \in \mathcal{E}_{\mathcal{N}^{(k)}(i)} \wedge \mathbf{y}_i = \mathbf{y}_j \mid j \in \mathcal{N}^{(k)}(i)\}|}{|\mathcal{E}_{\mathcal{N}^{(k)}(i)}|}, \quad (15)$$

where $\mathcal{N}^{(k)}(i)$ denotes the $k$-hop neighborhood of node $v_i$, and $(u, v) \in \mathcal{E}_{\mathcal{N}^{(k)}(i)}$ if and only if there exists a shortest path of length $k$ between $u$ and $v$ in the subgraph induced by $\mathcal{N}^{(k)}(i)$. The value $h_i^{(k)}$ represents the distribution within each node's $k$-hop neighborhood.

The results in Figure 7 indicate that the distributions vary significantly across different hops, highlighting the need for a decoupled graph prompting procedure. Each hop may carry unique and diverse information, contributing to the final representations to different degrees. Moreover, the distributions vary across different graphs, emphasizing the need for hop-specific approaches.

**Table 8: Summary of the mean and standard deviation of accuracy over all runs on few-shot node classification tasks. The best results for each dataset are highlighted in gray. GAT is used as the backbone encoder.**

| | **Texas** (0.11) | | **Cornell** (0.31) | | **Wisconsin** (0.20) | | **Chameleon** (0.20) | | **Squirrel** (0.22) | | **Arxiv-year** (0.22) | | **Cora** (0.81) | |
| --- | --- | --- | --- | --- | --- | --- | --- | --- | --- | --- | --- | --- | --- | --- |
| | 5-shot | 10-shot | 5-shot | 10-shot | 5-shot | 10-shot | 5-shot | 10-shot | 5-shot | 10-shot | 5-shot | 10-shot | 5-shot | 10-shot |
| Label Ratio | 13.6% | 27.3% | 13.6% | 27.3% | 9.9% | 19.9% | 1.09% | 2.19% | 0.48% | 0.96% | 0.0147% | 0.0295% | 1.29% | 2.58% |
| GAT | 52.22±4.22 | 53.81±5.80 | 61.36±4.34 | 64.36±3.58 | 54.35±3.45 | 53.40±3.14 | 28.53±3.24 | 30.09±3.60 | 21.35±2.34 | 23.47±2.98 | 21.38±1.30 | 23.75±2.31 | 62.14±1.08 | 65.86±1.37 |
| Finetune-LP | 50.41±5.80 | 54.89±7.13 | 60.72±3.58 | 65.72±3.97 | 56.36±3.19 | 52.13±2.78 | 30.24±3.10 | 33.58±2.89 | 23.45±1.37 | 23.87±2.45 | 22.15±2.34 | 25.36±2.80 | 65.47±1.35 | 68.37±2.01 |
| GPPT | 57.59±3.02 | 59.54±5.57 | 65.57±3.68 | 64.83±2.98 | 50.35±2.97 | 52.41±1.98 | 40.56±2.94 | 47.67±3.01 | 25.67±2.46 | 29.46±2.09 | 23.27±3.27 | 25.35±1.72 | 58.46±2.85 | 63.84±1.36 |
| Gprompt | 72.98±5.37 | 73.27±3.56 | 76.37±4.83 | 83.32±2.76 | 67.84±3.03 | 69.83±2.75 | 45.65±3.93 | 52.07±2.90 | 31.47±1.09 | 31.64±1.47 | 28.46±2.37 | 29.35±1.36 | 70.06±2.94 | 72.55±1.83 |
| GPF-LP | 68.42±2.39 | 70.28±4.29 | 72.12±4.58 | 78.34±3.85 | 62.47±3.84 | 67.45±3.53 | 40.61±6.04 | 41.56±4.24 | 29.46±1.75 | 30.15±3.73 | 28.10±3.65 | 28.34±2.31 | 68.04±1.86 | 69.74±1.46 |
| GPF-Plus-LP | 73.78±3.21 | 75.37±3.27 | 80.36±3.85 | 82.46±3.01 | 69.43±2.41 | 71.34±3.08 | 45.45±3.85 | 51.39±2.97 | 32.57±2.01 | 34.27±2.71 | 28.84±2.72 | 31.01±2.34 | 68.53±1.47 | 70.86±1.55 |
| All-In-One | 72.36±3.46 | 74.91±4.27 | 82.18±2.98 | 83.49±3.27 | 70.34±2.80 | 70.76±3.10 | 45.96±2.57 | 51.90±3.08 | 31.35±2.08 | 33.46±1.90 | 27.65±2.02 | 30.74±1.83 | 69.52±1.95 | 72.44±2.38 |
| HGPrompt | 72.53±2.96 | 73.41±3.87 | 76.45±3.84 | 81.35±3.56 | 66.53±2.54 | 69.47±3.98 | 44.47±3.01 | 46.76±3.41 | 30.37±1.65 | 32.40±2.42 | 28.41±2.64 | 30.01±2.55 | 67.75±2.31 | 71.96±1.94 |
| DAGPrompt | 78.27±3.03 | 82.25±2.61 | 86.05±1.77 | 88.73±1.99 | 76.55±1.22 | 77.22±2.47 | 49.98±2.31 | 54.11±1.96 | 34.80±1.42 | 37.03±2.17 | 29.37±2.84 | 32.56±1.62 | 70.48±1.41 | 73.18±0.82 |
| *Improvement* | 4.49 | 5.84 | 3.87 | 4.19 | 5.31 | 4.48 | 3.81 | 2.04 | 2.23 | 2.76 | 0.53 | 1.55 | 0.33 | 0.34 |

# E  Additional Experiments

## E.1  DAGPrompt with GAT as Backbone

To evaluate the generalization capability of DAGPrompt, we conduct 5-shot and 10-shot node classification experiments using GAT as the backbone encoder[2]. The results in Table 8 show that DAGPrompt consistently outperforms all baselines, providing strong evidence of its robust generalization performance.

## E.2  Full-shot Experiments

**Table 9: Summary of the mean and standard deviation of accuracy across all runs on full-shot node classification tasks. The best results for each dataset are highlighted in gray. GCN is used as the backbone model.**

| | **Chameleon** (0.20) | **Squirrel** (0.22) | **Cora** (0.81) |
| --- | --- | --- | --- |
| GCN | 48.57±2.76 | 38.67±2.10 | 78.57±1.23 |
| H2GCN | 69.54±1.24 | 57.31±2.47 | 77.65±0.84 |
| GPR-GNN | 70.22±1.45 | 58.19±3.40 | 79.56±0.87 |
| ALT-GNN | 70.51±1.23 | 59.34±1.38 | 78.45±1.85 |
| Finetune-LP | 50.73±2.45 | 39.61±1.98 | 76.67±1.37 |
| GPPT | 59.61±1.23 | 36.63±1.65 | 63.71±2.12 |
| GraphPrompt | 69.63±1.67 | 51.23±1.74 | 78.58±1.04 |
| GPF-LP | 61.49±2.53 | 50.09±2.08 | 74.16±1.47 |
| GPF-Plus-LP | 68.46±1.90 | 58.65±1.45 | 79.51±1.36 |
| All-In-One | 69.56±1.08 | 58.46±1.80 | 77.76±0.86 |
| HGPrompt | 68.50±2.48 | 53.65±1.97 | 78.15±0.74 |
| GCOPE | 70.03±1.35 | 56.24±2.03 | 80.09±1.85 |
| DAGPrompt | 74.30±0.78 | 62.78±1.29 | 81.10±0.72 |
| *Improvement* | 3.79 | 3.44 | 1.01 |

We further evaluate DAGPrompt under the full-shot setting, using a training-validation-test split of approximately 50%-25%-25%. The results in Table 9 show that, although the performance improvement is less pronounced compared to the few-shot settings in

[2]Baselines incompatible with GAT are excluded from these experiments. The number of heads is set to 4.

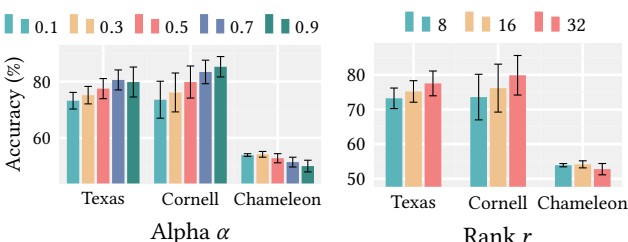

**Figure 8: Parameter analysis.**

Table 3, DAGPrompt consistently achieves the best results across all datasets, particularly on those with strong heterophily.

## E.3  Parameter Analysis

We conduct a parameter analysis for $\alpha$ and $r$ in Figure 8. The results show that Texas and Cornell generally perform better with larger $\alpha$ values, up to 0.9, while Chameleon favors smaller values, down to 0.1. This indicates that Texas and Cornell benefit more from distant hop information (as larger $\alpha$ assigns greater weight to them), whereas Chameleon relies more on local hop information with a smaller $\alpha$. Additionally, variations in the rank $r$ impact performance differently across datasets.

## E.4  Visualization of GLoRA Weights

We visualize the weight distributions of GLoRA on the Texas and Cornell datasets in Figure 9. For clarity, we present the final result of $A + P_A Q_A^\top$, representing the edge weights used during message passing in GNNs. As shown, GLoRA strengthens certain connections by assigning weights greater than 1, while weakening others with weights less than 1. This suggests that GLoRA refines the graph message-passing scheme to better align with downstream tasks.

## E.5  Pre-training Helps the Convergence

To examine the effect of pre-training on model convergence, we conduct experiments with varying hidden sizes of DAGPrompT on

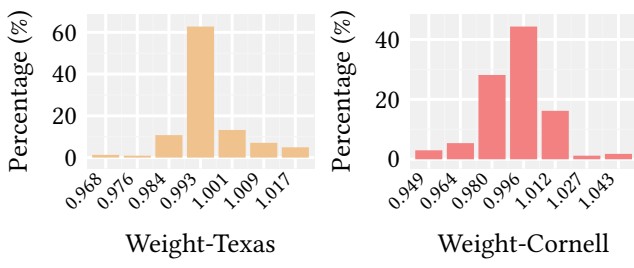

Figure 9: GLoRA weights distributions of $A + P_A Q_A^\top$

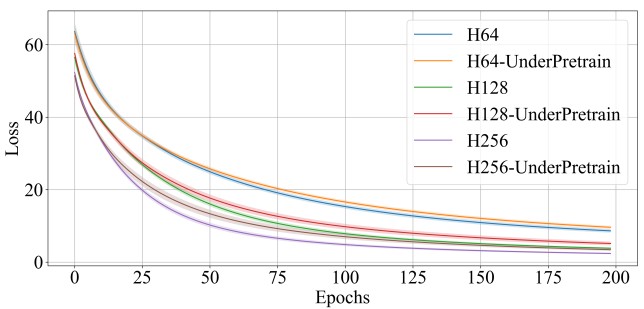

Figure 10: Loss curve of DAGPrompT on Chameleon.

the Chameleon dataset. For each configuration, the model is pre-trained either sufficiently (200 epochs) or minimally (30 epochs).

As shown in Figure 10, the results demonstrate that sufficient pre-training not only accelerates convergence (evident from a faster reduction in loss) but also improves final performance (achieving a lower loss). This aligns with findings in NLP fields, where sufficient pre-training reduces the intrinsic dimension of the model, simplifying the learning process for downstream tasks [1].

## E.6 DAGPrompT with Full-parameter Tuning

Table 10: Summary of the mean and standard deviation of accuracy across all runs for 5-shot node classification tasks.

|  | **Texas** (0.11) | **Wisconsin** (0.20) | **Squirrel** (0.22) |
|---|---|---|---|
| DAGPrompT | $81.36_{\pm4.93}$ | $76.37_{\pm1.17}$ | $53.38_{\pm1.97}$ |
| DAGPrompT-Full | $79.36_{\pm3.78}$ | $75.37_{\pm2.42}$ | $50.38_{\pm5.38}$ |
| DAGPrompT-Freeze | $79.01_{\pm4.29}$ | $74.74_{\pm4.37}$ | $50.61_{\pm4.70}$ |

We conducted an experiment with two variants of DAGPrompT: DAGPrompT-Full and DAGPrompT-Freeze. DAGPrompT-Full removes the GLoRA module and fine-tunes all GNN encoder parameters during prompting, while DAGPrompT-Freeze removes the GLoRA module and freezes all GNN encoder parameters during prompting. The results show that both DAGPrompT-Full and DAGPrompT-Freeze perform worse than DAGPrompT. This supports the theory from section 4 that neither zero-parameter nor full-parameter tuning is optimal in few-shot settings.

Received 20 February 2007; revised 12 March 2009; accepted 5 June 2009

