# OpenReview forum: "DAGPrompT: Pushing the Limits of Graph Prompting with a Distribution-aware Graph Prompt Tuning Approach"
_ACM.org/TheWebConf/2025/Conference — WWW 2025 Poster_

### Official Review · Reviewer_Jz67 · 2024-11-20

**Novelty:** 3
**Technical Quality:** 4

**Review:**

### 1. Summary
This work identifies two key challenges in graph prompt tuning for complex graphs, particularly heterophilic graphs. To address these challenges, it introduces **Distribution-Aware Graph Prompt Tuning (DAGPrompT)**, which integrates the **GLoRA module** and **hop-specific prompts**. The GLoRA module focuses on adjusting model parameters and enhancing message passing, while hop-specific prompts effectively incorporate multi-hop information to improve predictions. The study presents a clear motivation and comprehensive experiments, demonstrating the superiority of the proposed method over existing graph prompt tuning approaches.

### 2. Strengths
1. The challenges of adapting existing graph prompting methods to complex graphs are identified.
2. The experiments are extensive which show the superiority of this method compared to previous graph prompt tuning methods.
3. The writing is easy to understand.

### 3. Weaknesses
1. The proposed GLoRA has been extensively studied in prior research [1,2], exhibiting notable similarities to the primary contributions of this work, which limits its novelty. Furthermore, utilizing multi-hop information for prediction is relatively straightforward and lacks significant innovation [3].
2. GLoRA for message passing primarily modifies existing edges to improve efficiency. However, in heterophilic graphs, certain critical high-hop edges that do not exist in the original graph structure may hold greater importance [4], a limitation that the current GLoRA module fails to address effectively.
3. Graph prompt tuning is designed to optimize continuous prompts for simplifying downstream tasks. However, the primary contributions of this work largely involve adding additional parameters to adjust model configurations, which aligns more closely with variations of low-rank adaptation or adapter methods rather than graph prompt tuning.

**Questions:**

1. Recent advancements in graph contrastive learning methods have shown strong performance on both homophilic and heterophilic graphs [6,7,8]. In light of this, does the motivation for this work (Figure 1) remain compelling? These existing methods seem adept at distinguishing node embeddings across class disparities.
2. The experiments primarily focus on traditional GNN models such as GCN and GAT. How does the proposed method perform when applied to more advanced GNN architectures designed to handle both homophilic and heterophilic graphs, such as H2GCN [4] and AERO-GNN [5]?
3. Is the proposed method compatible with more advanced pretraining techniques, such as simGRACE [9] or MUSE [6], rather than simpler approaches like LP? Additionally, how sensitive is the method to different pretraining strategies, and how does its performance vary under these scenarios?

### References
[1] Li S, Han X, Bai J. Adaptergnn: Parameter-efficient fine-tuning improves generalization in gnns[C]//Proceedings of the AAAI Conference on Artificial Intelligence. 2024, 38(12): 13600-13608.

[2] Gui A, Ye J, Xiao H. G-adapter: Towards structure-aware parameter-efficient transfer learning for graph transformer networks[C]//Proceedings of the AAAI Conference on Artificial Intelligence. 2024, 38(11): 12226-12234.

[3] Abu-El-Haija S, Perozzi B, Kapoor A, et al. Mixhop: Higher-order graph convolutional architectures via sparsified neighborhood mixing[C]//international conference on machine learning. PMLR, 2019: 21-29.

[4] Zhu J, Yan Y, Zhao L, et al. Beyond homophily in graph neural networks: Current limitations and effective designs[J]. Advances in neural information processing systems, 2020, 33: 7793-7804.

[5] Lee S Y, Bu F, Yoo J, et al. Towards deep attention in graph neural networks: Problems and remedies[C]//International Conference on Machine Learning. PMLR, 2023: 18774-18795.

[6] Yuan M, Chen M, Li X. MUSE: Multi-View Contrastive Learning for Heterophilic Graphs[C]//Proceedings of the 32nd ACM International Conference on Information and Knowledge Management. 2023: 3094-3103.

[7] Chen J, Lei R, Wei Z. PolyGCL: GRAPH CONTRASTIVE LEARNING via Learnable Spectral Polynomial Filters[C]//The Twelfth International Conference on Learning Representations. 2024.

[8] Wan G, Tian Y, Huang W, et al. S3GCL: Spectral, Swift, Spatial Graph Contrastive Learning[C]//Forty-first International Conference on Machine Learning.

[9] Xia J, Wu L, Chen J, et al. Simgrace: A simple framework for graph contrastive learning without data augmentation[C]//Proceedings of the ACM Web Conference 2022. 2022: 1070-1079.

**Reviewer Confidence:**

4: The reviewer is certain that the evaluation is correct and very familiar with the relevant literature

**Scope:**

4: The work is relevant to the Web and to the track, and is of broad interest to the community

---

### Official Review · Reviewer_URyD · 2024-11-24

**Novelty:** 4
**Technical Quality:** 5

**Review:**

This claim introduces a interesting method (DAGPrompT) and its specific components (GLoRA and low-rank adaptation) that likely have established theories or previous implementations. References are crucial to validate the methodology and its effectiveness in addressing the identified challenges. This paper gives a relatively complete experimental discussion process and brief theoretical analysis. But its novelty is limited as the work faces up with some drawbacks and weaknesses.

**Questions:**

1 There are misconceptions in this work, which is to use existing concepts as one's own innovative point. For example, the concept of hop-decoupled manner is common and easy in the Graph-based Deep Learning. The issue of different degrees of homology among neighbors of different hops has also been studied by researchers for a long time like [1].
[1]Du, L., Shi, X., Fu, Q., Ma, X., Liu, H., Han, S., & Zhang, D. (2022). Gbk-gnn: Gated bi-kernel graph neural networks for modeling both homophily and heterophily. In Proceedings of the ACM Web Conference 2022 (pp. 1550-1558).
2 You statemented the model can be applied to Link-Prediction task in subsection 3.2, but I can not find the corresponding results in Experiments part. You should presents the comparison with SOTA link prediction baselines.
3 From Figure. 4 we can observe some interesting results, for example, models perform worst around 0.2 related to homophily ratio. It can be observed that from Table 3 that the  homophily ratio of ‘Chameleon’ is 0.2. I am wondering that weather the raw topology is worst for the dateset or not. Please give more explanations to clarify the reasons.
4 In lines. 355-358, the sentence ‘A theoretical analysis is offered in section 4, with experimental results in subsection E.6.’ appears two times.

**Reviewer Confidence:**

4: The reviewer is certain that the evaluation is correct and very familiar with the relevant literature

**Scope:**

4: The work is relevant to the Web and to the track, and is of broad interest to the community

---

### Official Review · Reviewer_RSvb · 2024-12-01

**Novelty:** 5
**Technical Quality:** 5

**Review:**

**Summary**:\
The paper proposes DAGPrompT, which optimizes GNNs for heterophily graphs using GLoRA and hop-specific prompts.

**Strengths**:
- Designing prompt learning methods for heterophily graphs is a significant and underexplored research area.
- The authors conduct extensive experiments to demonstrate the effectiveness of the proposed method.
- The authors provide open-source code, ensuring reproducibility and transparency.

**Weaknesses**:
-  Using *"distribution"* to describe the nature of heterophily graphs is uncommon and may be confusing. This terminology could be clarified or polished for better understanding.
- [1,2] are highly related baselines that should be compared and discussed to contextualize the contributions of this work.
- Some graph prompting methods already consider hop differences by employing different prompts to modify the input features and each hidden layer's output [2,3]. This contradicts the claim that *"Existing methods typically rely on the final layer representation of the GNN encoder, overlooking these hop-specific variations."*

[1] Yu et al. Non-homophilic graph pre-training and prompt learning. KDD 2025.\
[2] Yu et al. Generalized graph prompt: Toward a unification of pre-training and downstream tasks on graphs. TKDE 2024.\
[3] Yu et al. MultiGPrompt for multi-task pre-training and prompting on graphs. WWW 2024.

**Questions:**

I am open to adjusting my score based on the authors' responses.

**Reviewer Confidence:**

3: The reviewer is confident but not certain that the evaluation is correct

**Scope:**

4: The work is relevant to the Web and to the track, and is of broad interest to the community

---

### Official Review · Reviewer_CyyA · 2024-12-01

**Novelty:** 5
**Technical Quality:** 4

**Review:**

This work proposes a distribution-aware graph prompt tuning approach, which distinguishes itself as a pioneering model in extending the capabilities of graph prompting techniques to heterophily graphs.

The strengths can be summarized as below,

- Innovative Methodology:  The integration of GLoRA for low-rank adaptation enables effective tuning of pre-trained GNNs while preserving computational efficiency.
The Hop-specific Graph Prompting module provides fine-grained adaptability to hop-specific variations, a notable improvement over traditional methods.

- Comprehensive Evaluation:  Experiments conducted on 10 datasets, including both node and graph classification tasks, highlight the robustness and versatility of the approach.
The model achieves up to 7.55% improvement in accuracy on complex heterophilic graphs, showcasing state-of-the-art performance.

- Tackling Critical Challenges: The paper identifies and resolves key limitations in existing prompting methods, such as the inability to adapt to heterophilic graphs and hop-specific requirements. The analysis of the pre-training/fine-tuning gap and its mitigation is insightful and well-grounded.

**Weaknesses:**
- Generalization Beyond Datasets: While the method performs well on the selected datasets, its applicability to other domains or extremely large-scale graphs is not fully explored.
- Efficiency Metrics: Although efficiency is discussed, a deeper comparative analysis against other lightweight approaches could strengthen the claims.
- Complexity analysis: The addition of multiple components (GLoRA, hop-specific prompts) increases model complexity. While beneficial, this might hinder real-time applications or scalability without further optimization.

**Questions:**

Q1. How the authors select the parameters for fine-tuning?

Q2. Some literature and discussions of related works are not sufficient, e.g., GReTo [1] proposes an adaptive topology optimization for  graphs with both homophily and heterophily  components.

Q3-Q5: Please see my listed weaknesses, regarding generalization on other domains (more discussions), efficiency metrics, as well as more complexity analysis.


[1] Zhou Z, Huang Q, Lin G, et al. Greto: Remedying dynamic graph topology-task discordance via target homophily[C]//The eleventh international conference on learning representations. 2023.

**Reviewer Confidence:**

3: The reviewer is confident but not certain that the evaluation is correct

**Scope:**

3: The work is somewhat relevant to the Web and to the track, and is of narrow interest to a sub-community

---

### Official Review · Reviewer_YYMb · 2024-12-03

**Novelty:** 5
**Technical Quality:** 6

**Review:**

Pre-training and adopting to downstream tasks via prompt tuning is a common paradigm for few-shot learning in graphs. In the paper, the authors identify two challenges namely i) discrepancy between the pre-training task and the downstream task especially in the heterophily graphs and ii) customising prompts for hop-specific node requirements. To overcome these challenges, the authors propose Distribution Aware Graph Prompt Tuning (DAGPrompT) which integrates Graph Low Rank Adaptation (GLoRA) and layer-wise prompting. The authors show that the proposed approach archives better performance in few-shot node and graph classification tasks as compared to existing baselines, especially in datasets with strong heterophily.

Strengths:

i) The paper is well written and easy to follow

ii) The research gap is clearly highlighted, namely existing prompt based methods are good few shot learners but does not account for heterophily whereas existing hetrophily methods cannot effectively learn with few labelled samples. The authors propose prompt based method for few shot learning capable of handling both heterophily and homophily graphs.

iii) Using layer-wise prompts to take advantage of different hops

iv) Extensive experiments showcase the efficacy of the proposed method in hetrophily graphs

Concern:

i) It would be helpful if the authors provide an explanation for how GLoRA is different from LoRA other than using $P_AQ_A^T$

**Questions:**

Questions:

i) The authors showcase the efficacy of GLoRA in ablation study. In addition to that, showcasing the effect of not using $P_AQ_A^T$ would be better

ii) Does w/o layer-specific prompts in ablation study mean only last layer is used for prompts?

iii) How $\tilde{S}$ is used in eq 10 is not clear

**Reviewer Confidence:**

3: The reviewer is confident but not certain that the evaluation is correct

**Scope:**

4: The work is relevant to the Web and to the track, and is of broad interest to the community